# Next-Generation Boron Drugs and Rational Translational Studies Driving the Revival of BNCT

**DOI:** 10.3390/cells12101398

**Published:** 2023-05-16

**Authors:** Danushka S. Seneviratne, Omran Saifi, Yuri Mackeyev, Timothy Malouff, Sunil Krishnan

**Affiliations:** 1Department of Radiation Oncology, Mayo Clinic Florida, Jacksonville, FL 32224, USA; 2Department of Neurosurgery, UTHealth, Houston, TX 77030, USA; 3Department of Radiation Oncology, University of Oklahoma, Oklahoma City, OK 73019, USA

**Keywords:** BNCT, High-LET, immunomodulation, novel boron compounds

## Abstract

BNCT is a high-linear-energy transfer therapy that facilitates tumor-directed radiation delivery while largely sparing adjacent normal tissues through the biological targeting of boron compounds to tumor cells. Tumor-specific accumulation of boron with limited accretion in normal cells is the crux of successful BNCT delivery. Given this, developing novel boronated compounds with high selectivity, ease of delivery, and large boron payloads remains an area of active investigation. Furthermore, there is growing interest in exploring the immunogenic potential of BNCT. In this review, we discuss the basic radiobiological and physical aspects of BNCT, traditional and next-generation boron compounds, as well as translational studies exploring the clinical applicability of BNCT. Additionally, we delve into the immunomodulatory potential of BNCT in the era of novel boron agents and examine innovative avenues for exploiting the immunogenicity of BNCT to improve outcomes in difficult-to-treat malignancies.

## 1. General Overview of BNCT

One of the most challenging aspects of oncological care is improving the therapeutic ratio of treatment by increasing cancer cell death while limiting the probability of normal tissue complications. Boron neutron capture (BNCT) is a biologically targeted radiation treatment modality designed to improve tumor control while reducing damage to surrounding tissues. BNCT is classified as a high-linear-energy transfer (high-LET) therapy and is distinguishable from low-LET photon and proton therapy given its ability to form clustered, direct DNA damage in a less oxygen-dependent manner. BNCT involves the selective delivery of boronated compounds into tumor cells, followed by exposure of the tumor to neutron irradiation. The nonradioactive boron-10 atoms (^10^B) then react with thermalized neutrons via a nuclear capture and fission reaction, leading to the production of a high-LET, low-energy, alpha particle and a recoil lithium-7 atom (^10^B + ^1^n → [^11^B]* → ^4^He + ^7^Li) [1,2,3]. The high-LET alpha particles generated from this reaction form dense ionization tracks along cellular macromolecules. In comparison to low-LET radiation, the formation of clustered, dense ionization tracks along DNA produces more complex damage characterized by co-localization of different forms of DNA damage—base excision, single-strand breaks, and, the most damaging, double-strand breaks. Preclinical studies have demonstrated that the densely clustered DNA damage (two or more DNA lesions within one turn of DNA) formed by high-LET therapy is more difficult to repair and is associated with increased genomic instability and greater tumor cell death. Furthermore, given that the alpha particles generated through BNCT have a range of <10 μm, the resulting DNA damage only occurs within the diameter of a single tumor cell, sparing adjacent normal tissues [4]. Historically, neutron sources for BNCT have primarily been based on fission reactors, as such reactors were able to produce the appropriate neutron flux necessary for BNCT. These use neutrons that arise directly from the core following filtration and moderation of the beam. The early reactor-based BNCT facility at Brookhaven National Laboratory during the 1950s–1960s largely produced thermal neutrons. In order to produce a neutron beam that is useful in the treatment of deep-seated tumors, however, the thermal and fast components of the beam must be reduced, while the epithermal component must be enhanced. Given this, investigations into increasing the epithermal neutron proportion ensued, and fission-reactor-based neutron production was further improved upon to ensure the necessary beam intensity for clinical treatments [5]. Although the historical importance of nuclear reactors in the development of clinical BNCT cannot be understated, given their large size and their expensive nature, clinical BNCT is largely performed at accelerator-based facilities. Nevertheless, there are several reactor-based BNCT sources that still remain operational in Argentina, China, Japan, and Taiwan. Accelerator-based neutron sources can be easily installed within a hospital environment and are more cost-effective for clinical use. In accelerator-based BNCT, fast neutrons are produced by bombarding lithium or beryllium targets with protons. These fast neutrons, however, cannot be utilized for BNCT and are moderated by the beam-shaping assembly to result in a collimated beam consisting primarily of epithermal neutrons [5,6]. A schematic overview of BNCT is demonstrated in Figure 1.

The ideal boron agent should selectively accumulate within tumor cells, demonstrate low systemic effects, and be cleared from normal tissues and blood following treatment delivery. Additionally, the ^10^B load per cell should be maximized with at least 20 μg of ^10^B per gram of tumor [7]. First-generation boron compounds including boric acid and its derivatives were originally developed during the 1950s–1960s. Given their low tumor selectivity and high toxicity profiles, these were discontinued from clinical use [1,7]. Second-generation boron compounds include sodium borocaptate (BSH) and (L)-4-boronophenylalanine. Clinical BNCT trials to date have primarily utilized second-generation compounds [7,8]. BSH entry into tumor cells occurs via passive diffusion and relies solely on the enhanced permeability and retention (EPR) effect whereby tumors with leaky, chaotic, and immature neovasculature with large inter-endothelial fenestrations and underdeveloped basement membranes allow passive extravasation of macromolecules into the tumor interstitium, and the poor lymphatic drainage from these tumors sequesters these extravasated macromolecules within the tumor even longer. Despite its low tumor accumulation and poor blood–brain barrier penetration, BSH has shown success in multiple preclinical models and has been clinically utilized in the treatment of a variety of tumors, including glioblastoma and recurrent head and neck cancers. Malouff and colleagues have published a relatively comprehensive review of clinical BNCT involving the use of BSH vs. BPA [3]. This is attributed to its ability to carry large amounts of boron-10 and its low systemic toxicity [9,10,11]. BPA is commonly combined with fructose to develop F-BPA, as this improves solubility and allows for higher tumor boron delivery [1]. F-BPA enters cells via the L-amino acid transporter 1 (LAT-1) amino acid transporter. LAT-1 is overexpressed in metabolically active cells and is highly upregulated in a variety of cancer types [1,12]. BPA shows tumor selectivity and low toxicity and has demonstrated considerable promise as a boron carrier in multiple clinical trials involving glioblastoma, meningioma, head and neck carcinoma, and melanoma [1,3]. Interestingly, Nomoto and colleagues recently reported that poly(vinyl alcohol) (PVA) can form complexes with BPA and undergo internalization within cancer cells through LAT1-mediated endocytosis. Animal studies performed using this combination compound demonstrated enhanced anti-tumor effects, suggesting that LAT-1-mediated boron entry could be further manipulated to improve tumor outcomes [13].

An ongoing challenge associated with BNCT involves accurately calculating the dose delivered to tumor and normal tissues. The concept of relative biological effectiveness (RBE) is used to describe the efficacy of other types of high-LET radiation and is often employed to assist in adjusted dose calculations. RBE is defined as the ratio of the absorbed dose from a given radiation modality to a reference type of low-LET radiation (such as X-rays) needed to achieve a predefined biological impact. Although the biological effectiveness of BNCT is likely greater than that of low-LET radiation, it is important to note that the concept of RBE may only be accurately applied when the quantity of the absorbed dose can be clearly defined, as in the case of traditional high-LET therapies such as carbon and helium ion therapy. Given that boron distribution in cells during BNCT administration is often inhomogeneous, the concept of RBE cannot solely be used to estimate dose. A more appropriate method of describing the effectiveness of BNCT is using a concept known as compound biological effectiveness (CBE), which is defined as the product of RBE and cellular boron distribution. CBE is dependent on the physical and chemical properties of the boronated compound, the means of administration, and the microdistribution of boron within tumor and normal tissues. [14,15,16]. Fukuda and colleagues provide a comprehensive review of the literature relating to RBE and CBE values of the two most commonly utilized boron compounds, BPA and BSH, with regard to normal tissues such as skin, central nervous system, lung, mucosal tissues, and liver [14]. Hsu and colleagues reported on the microdosimetry of the THOR BNCT beam utilizing two tissue-equivalent proportional counters and reported that neutron RBE, photon RBE, and boron capture RBE within the phantom vary significantly at differing depths [15].

To accurately predict the delivered dose, CBE must be determined experimentally for each boron compound, as well as for various malignant and normal tissues [16,17,18]. Ono describes the impact of the boron compound structure on CBE for both BSH and BPA and notes that the identification of CBE for various compounds and cell types will accelerate the progression of BNCT research [18]. Masunaga and colleagues examined the CBE of BPA and BSH in mouse xenografted tumors and determined that with increasing concentrations of each boron agent, the CBE value decreased; however, this effect was more pronounced for BPA. They also speculated that tumor characteristics such as micro-environment, heterogeneity, and genetic factors are likely to partially influence CBE [19]. As it would be nearly impossible to assess CBE for every compound in every tumor type, assumptions based on available BNCT data in humans and preclinical models are often used to make estimates regarding the RBE and CBE. Nonetheless, as the delivered radiation dose may differ based on the compound used for boron delivery and the treated disease site, devising an accurate model to estimate dose during BNCT is of paramount importance and remains an area of active investigation [17,18,19].

Until recently, the use of BNCT has been limited to small Phase I/II trials in a variety of disease sites, including primary and recurrent head and neck cancer, glioblastoma, melanoma, meningioma, sarcoma, hepatocellular carcinoma, and malignant mesothelioma [3,20,21]. However, during May of 2020, BNCT was authorized to be covered under the National Health Insurance system in Japan for the treatment of recurrent head and neck cancers. The boron compound approved for this treatment was Borofalan [10B] (otherwise known as Steboronine^®^) [22]. The clinical approval of BNCT in this setting was fueled by the results of the Phase II JHN002 clinical trial, in which patients with recurrent or locally advanced non-squamous cell carcinoma of the head and neck region were administered Borofalan (^10^B) and treated with neutron irradiation. The primary endpoint of the JHN002 trial was the objective response rate (ORR). The ORR for all patients was 71%. A complete response rate of 50% and a partial response rate of 25% were noted in those with recurrent disease, and a complete response rate of 8% and a partial response rate of 62% were observed in locally advanced patients. The 2-year overall survival was 58% for recurrent patients vs. 100% for those with locally advanced disease [21]. Borofalan (^10^B) for this trial was provided by Stella Pharma, who continue to support ongoing clinical trials assessing the use of Borofalan in malignant melanoma and hemangiosarcoma.

Despite this recent triumph and the potential therapeutic advantages of BNCT, the widespread adoption of this modality has been hindered by several factors. These include the limited availability of therapeutic epithermal neutron beams, difficulty in identifying non-toxic tumor-selective boronated compounds, and uncertainties related to BNCT dosimetry and treatment planning [2]. However, given the growing national and global interest in improving the therapeutic efficacy of oncologic therapies, preclinical and clinical investigations related to BNCT have significantly expanded over the last decade.

## 2. Unique Advantages of BNCT

In comparison to low-LET modalities, the direct, dense ionization tracks created following BNCT are likely to lead to increased tumor cell death, even when faced with traditionally radioresistant tumors and hypoxic tumor microenvironments. Its biologically targeted nature is expected to spare adjacent normal tissues to a greater degree than X-rays and therefore is likely to permit greater dose escalation within tumors. These unique aspects of BNCT are expected to improve local tumor control while limiting the toxicities associated with treatment [2,20,23]. Additionally, given that cancer often tends to be a systemic disease, there is increasing interest in understanding and exploiting the systemic impacts of BNCT therapy. These include the potential to induce abscopal effects at distant sites and the concept of converting immunologically “cold” tumors to “hot” ones to improve their response to immunomodulatory agents. Albeit weak, a wide variety of immune-related cellular and molecular changes have been observed following X-ray therapy [24]. Preclinical and limited clinical works now suggest that high-LET therapies such as carbon ions and BNCT are highly immunogenic and have the capability to induce greater immunomodulatory impacts on tumors. The potential to synergize systemic effects of BNCT with targeted therapies and immunomodulatory drugs is of tremendous interest to the oncological community [25,26,27].

In this review, we will delve into investigations regarding the development of novel boron carriers and discuss the existing translational research on the biological impacts of BNCT, its proposed immunomodulatory impacts, and the potential for synergizing BNCT with other cancer-directed therapies.

## 3. Investigations into Novel Boron Carriers

As neither BPA nor BSH are ideal boron carriers, there has been immense interest in developing a third generation of boronated compounds to increase selective tumor boron uptake while limiting normal tissue and systemic toxicities. The majority of these efforts have focused on integrating boron with tumor-specific entities such as antibodies, peptides, polyamines, liposomes, and nanoparticles [7]. Several novel boron carriers and their mechanisms of boron delivery are outlined in Figure 2. Outlined below is a non-exhaustive summary of promising paradigms for boron delivery to tumors that go beyond the use of BPA and BSH.

One method of boron delivery involves the use of peptide conjugates consisting of boron clusters. Kimura and colleagues developed an icosahedral *o*-carborane boron cluster Arg-Gly-Asp (RGD) with a peptide conjugate to recognize Integrin α_v_β_3_ and α_v_β_5_. As Integrin α_v_β_3_ and α_v_β_5_ are highly expressed in endothelial cells and various types of tumor cells, RGD served as a tumor-targeting moiety. As a result, these boron clusters demonstrated increased tumor uptake and longer tumor retention in comparison to BSH [28]. In another study, cell-penetrating peptides were fused to BSH in order to improve BSH uptake through the cell membrane. Fusion of 8BSH to 11R penetrating peptide with a dendritic lysine structure allowed for localization of BSH to the nucleus, and led to increased cell death in glioma cells and mouse tumor models [29].

Using a similar boron cluster in recent work, the cyclic RGD (cRGD) peptide, known to bind to α_v_β_3_ and α_v_β_5_ integrins, was combined with a maleimide-functionalized *closo*-dodecaborate (MID) and bovine serum albumin (BSA) through covalent bonding to create cRGD-MID-BSA [30]. Upon neutron irradiation, significant tumor growth suppression was observed in subcutaneous U87MG malignant glioma xenografts at a cRGD-MID-BSA dose of 7.5 mg [^10^B]/kg given intravenously. The improved biodistribution with higher uptake in integrin-overexpressing U87MG tumors and a greater anti-tumor effect upon neutron irradiation was attributed to active targeting with cRGD since these effects were more pronounced than just MID-BSA. Incorporating boron into receptor-specific antibody carriers is a highly promising method of improving the tumor selectivity of BNCT. Early work on this concept was carried out by Wu and colleagues who conjugated the boron-containing polyamidoamine dendrimer G5-B1100 to the anti-EGFR monoclonal antibody Cetuximab. Intracerebral injection of this boron entity in rats bearing EGFR overexpressing gliomas led to the tumor-specific accumulation of the boron dendrimers [31]. More recently, this concept was improved upon by Nakase and colleagues, who described an antibody-based receptor-targeting boron compound with a novel Fc-binding peptide conjugate. As this conjugate is capable of interacting with the Fc domain of human IgG, it serves as a linker to which nearly any receptor-targeting antibody may be attached. As a proof of concept, the authors demonstrated that the conjugate can be linked to the heavily boronated compound dodecaborate. This was then bound to the Fc component of Cetuximab, a humanized chimeric monoclonal antibody targeting the epidermal growth factor receptor (EGFR), in order to target F98 rat glioma tumor cells overexpressing EGFR [32]. Similarly, monoclonal antibodies against the CD133 stem cell marker with attached boron conjugates were also shown to deliver high boron levels to rat glioma cells [33].

Boron-containing porphyrin compounds have been studied as potential drugs for BNCT given that porphyrins have low systemic toxicity and tend to naturally aggregate within malignant cells. Prototypical compounds include a tetrakis-carboxylate ester of 2,4-bis (α, β-dihydroxyethyl) deuterioporphyrin IX (also known as BOPP, a boronated protoporphyrin) and octa-anionic 5,10,15,20-tetra [3,5-(nido-carboranylmethyl)phenyl] porphyrin (H_2_OCP) [7,34]. Although these entities deliver large boron payloads, animal studies suggest that they have low tumor specificity. For instance, in F98 rat glioma models injected with boronated porphyrin compounds, boron accumulation was primarily noted in macrophages [35]. A group of 5,10,15,20-tetra[3,5-(carboranylmethyl)phenyl]porphyrins bearing eight *nido*-carborane cages (16 boron clusters) covalently attached to a porphyrin macrocycle, having 35–45% boron by weight, have also shown promise in BNCT [36]. Despite the high-boron cluster payload, some of these derivatives interact with DNA and thereby produce in vitro DNA damage [37,38]. In another study, Ozawa and colleagues attempted to assess tumor boron accumulation following direct BOPP delivery vs. intravenous injection. They found that tumor concentrations of the drug were much greater following direct intracerebral delivery in 9L rat glioma models. Admittedly, however, this is quite invasive and difficult to replicate in patients. An alternative approach to overcome the challenge of poor blood–brain barrier perfusion of porphyrins is to use convection-enhanced delivery to gliomas. Conversely, however, boronated porphyrins could serve as dual BNCT and PDT agents. Therefore, further research is needed to improve the delivery and tumor specificity of boron-containing porphyrin compounds [39].

Liposomes are vesicles surrounded by a lipid bilayer. Boron compounds may be conjugated to lipids to create boron-loaded liposomes. Alternatively, liposomes may also be used to encapsulate boronated compounds [7]. Boron liposomes targeted at human epidermal growth factor receptor 2 (HER-2) demonstrated receptor-specific binding in SK-BR-2 breast cancer cells and delivered high intracellular boron concentrations [40]. In a separate study, the boron compound Na_3_[1–(2′-B_10_H_9_)-2-NH_3_B_10_H_8_] was incorporated into liposomes and injected into mice bearing EMT6 cell mammary tumors, which led to significant tumor regression following BNCT treatment [41,42]. In addition to potentially improving targeted boron delivery to tumors, liposomes may also be utilized to enhance the immunogenicity of BNCT. This concept will be further discussed in the sections below.

Polymeric boron-containing nanoparticles can serve as robust boron delivery agents for BNCT. Nanoparticles are particularly attractive given their biological stability, selective accumulation within tumors, and their ability to deliver large amounts of boron [7,43,44]. Preclinical studies demonstrate that gold nanoparticles (AuNPs) may be successfully linked to boron to create AuNP-boron assemblies. Injection of AuNP-boron cage assemblies into mice bearing gastric cancer xenografts demonstrated selective boron accumulation within tumors, indicating that nanoparticles are a promising method of boron transport for BNCT [44]. Nanoparticle boron delivery may also be combined with other methods, including liposomal encapsulation. Recently, Singh and colleagues developed a liposome-based pure boron nanoparticle consisting of a polyethylene glycol surface, and a core containing a near-infrared fluorescent dye and boron nanoparticles. They then conjugated this entity to a tumor-specific ligand, folic acid. The liposomes had a diameter of 100–120 nm. The tumor targeting nature of folic acid was evidenced by the fact that folic-conjugated liposomes accumulated to a significantly greater degree in C6 brain tumor cells in comparison to liposomes lacking the conjugate. The level of intracellular boron accumulation through this method was confirmed to be adequate for therapeutic BNCT [45].

A different group of *o*-carborane derivatives, containing benzenesulfonamide and hydroxamic acid groups, prepared by using the “click” 1,3-dipolar cycloaddition reaction, was evaluated as matrix metalloproteinases (MMP) inhibitors [46]. MMPs are calcium- and zinc-dependent endopeptidases which function to degrade the extracellular matrix to allow tumor cells to invade and migrate efficiently; elevated levels of MMP-2, MMP-9, and MMP-13 have been identified and observed in cancer patients. On clonogenic survival analysis with neutron irradiation, the *o*-carborane derivatives had D_37_ (dose used to inhibit 63% colony formation) values of 0.27–0.32 Gy, compared to 0.82 Gy for BPA, and 1.55 Gy for boron-free control, suggesting that they may be promising BNCT agents too.

Among the most studied boron structures is boron nitride, a lattice-like crystalline structure of boron and nitrogen that is highly resistant to thermal or chemical denaturation. The hexagonal form is analogous to graphite but highly insoluble in water. Nanostructured hexagonal boron nitride is synthesized by heating a mixture of boric acid and ammonia in a stainless steel autoclave at a high temperature and pressure and has been shown to be highly water dispersible, allowing its use in biological applications [47].

Similarly, boron nitride nanotubes solubilized in water using 1,2-distearoyl-sn-glycero-3-phosphoethanolamine-poly(ethylene glycol) (DSPE-PEG2000) showed potent antitumor effects in in vitro studies of B16 melanoma cells treated with thermal neutrons [48]. A persisting challenge for in vivo applications is the decomposition in circulation that limits tumor delivery via intravenous delivery routes. An interesting solution to this challenge was to coat boron nitride nanoparticles with phase-transitioned lysozyme that protects the particle from hydrolysis, prolongs circulation time, and releases boron on demand within the tumor when ascorbic acid is infused intravenously. In 4T1 triple-negative breast cancer models, this strategy of controlled degradation and release of boron coupled with neutron irradiation resulted in a significant reduction of tumor volume in mice and 100% survival [49]. In a similar vein, boron carbide is a high boron-containing material with a twelve-boron icosahedron that lattices with carbon in a rhombohedral pattern with each carbon atom bridging three icosahedra. Surface functionalization of boron carbides with hyperbranched polyglycerol through ring-opening polymerization results in a hydrophilic construct that accumulates in tumors upon intravenous administration. Subsequent treatment with neutron irradiation or dual therapy (BNCT and photothermal therapy) with near-infrared laser illumination results in dramatic reductions in tumor volume. Given some of the promise of these studies, further investigation of optimized formulations of boron nitrides and carbides as BNCT agents is warranted [50].

## 4. Translational Work to Date Involving BNCT

Translational studies involving BNCT are limited. Here, we attempt to provide a comprehensive summary of the significant works published to date.

One of the earliest animal models in which BNCT was assessed was the hamster cheek pouch model for oral carcinoma. In this study, tumors mimicking human oral mucosal carcinoma were induced in hamsters, using dimethyl-1,2-benzanthracene as a carcinogen. Biodistribution and pharmacokinetic studies were then performed at various time points. Following administration of 300 mg BPA/kg, the mean boron uptake of tumor vs. normal pouch tissue was found to be 2.4 to 1, while the ratio of tumor vs. blood was 3.2 to 1. Increasing the injected dose of BPA increased tumor boron uptake [51]. Following verifying the pharmacokinetics of BPA, the authors then went on to assess the tumor response to BPA-based BNCT in this animal model. BNCT was performed using the epithermal beam of an RA-6 reactor at a prescription dose of 5 Gy absorbed by the tumor. At 15 days post-treatment, complete tumor remission and partial remission were observed in 78% and 13% of the animals, respectively. Normal tissue damage was found to be minimal [52]. These works eventually paved the path for the first human clinical trials of BNCT in recurrent head and neck cancer [8].

In an attempt to improve the therapeutic efficacy of BNCT, Ono and colleagues assessed the impact of BNCT delivery using a combination of BPA and BSH in C3H/He tumor-bearing mice. They noted that the BNCT delivered using this combined approach demonstrates improved tumor control in comparison to BNCT performed using individual compounds [53]. This combination approach has also demonstrated promising clinical results among malignant glioma patients, with a mean tumor reduction rate of approximately 50% [54]. In a similar vein, Schwint and colleagues then went on to investigate a novel boron delivery agent known as decahydrodecaborate (GB-10). When GB-10 was injected as a single agent, biodistribution studies failed to demonstrate selective accumulation of the compound in tumors. However, to the surprise of the investigators, it nevertheless led to selective tumor cell death in the hamster cheek pouch carcinoma model, with minimal impact on surrounding normal tissues. The authors note that the milder side effects of GB-10 on surrounding tissues in comparison to BPA permitted further dose escalation. It was postulated that the high in vivo tumor cell death observed when utilizing this agent despite its low tumor accumulation was primarily driven by its impact on aberrant tumor vasculature [55,56]. These studies demonstrated that accumulation of GB-10 within the abnormal tumor blood vessels can help selectively target tumors while sparing normal tissues and that high-boron payloads within tumor cells are not always necessary for the success of BNCT. When comparing GB-10 or similar vascular targeting agents to traditional boron carriers such as BPA, it is important to recognize that BPA is taken up by the LAT-1 transporter which is overexpressed in metabolically active cells [12]. Therefore, BPA can accumulate in metabolically active non-malignant tissues as well and has the potential to generate more normal tissue toxicity. Additionally, BPA can demonstrate heterogenous accumulation in tumors based on the metabolic activity of individual tumor cells, making it difficult to calculate the delivered dose. Ultimately, despite the high boron concentrations within tumors, these factors can limit the effectiveness of BPA-based BNCT, and approaches to overcome these limitations must be investigated.

Schwint and colleagues next combined GB-10 and BPA to determine whether the therapeutic impact of BNCT can be enhanced using dual methods of boron delivery. At an 8 Gy absorbed dose, the combination-therapy-based BNCT led to an approximately 93% tumor response rate (complete and partial response), with less mucositis in comparison to BPA-BNCT. The authors hypothesized that the decreased mucositis observed with combination therapy is the result of using less BPA, which likely limited damage to the metabolically active basal cell layer within the oral mucosa. The combination therapy also improved boron accumulation homogeneity within the tumor [57]. These studies demonstrated that the ideal boron delivery agent may in fact involve a combination of drugs that work through differential mechanisms.

To further improve outcomes using combination therapy, a sequential BNCT delivery method separated by 24–48 h was then investigated. This concept was based on the theory that the first BNCT application using BPA-BNCT would lead to cell death, and therefore reduce the interstitial fluid pressure within the tumor. The reduction in fluid pressure would then allow for the increased intratumoral accumulation of GB-10. The dual treatment was also predicted to target cell populations that did not respond to the initial round of BPA-BNCT. As projected, the tumor response increased from 75% with a single application to 90% with dual sequential applications [58]. This work suggested that temporal separation of BNCT delivery and the use of multiple boron agents may help improve patient treatment outcomes following BNCT.

In order to improve the effectiveness of BNCT, it is vital to understand the mechanisms underlying BNCT-induced cell death. Rodriguez and colleagues investigated the patterns of DNA damage and cell death in a thyroid carcinoma cell line following BNCT. The presence of γH2AX foci and the expression of the main effectors of homologous recombination and non-homologous end-joining pathways, including Ku70, Rad51, and Rad54, were assessed. This work demonstrated that the primary DNA double-strand break (DSB) repair pathway activated following BNCT was homologous recombination repair [59]. In a related study assessing boric-acid-mediated BNCT in a hepatocellular carcinoma (HCC) cell line, DNA damage induced via BNCT was again noted to be primarily repaired through the homologous recombination repair pathway. BNCT also led to G2/M cell cycle arrest and apoptosis-mediated cell death in the HCC cell line. These works suggest that combining BNCT with inhibitors of homologous recombination repair may yield increased tumor regression, potentially improving the efficacy of BNCT.

In another study, carborane-derived liposomes, known as boronsomes, were used as boron carriers, and the effectiveness of BNCT utilizing this method was investigated in mammary-tumor-bearing mice [60]. Boronsomes were generated by covalently attaching carboranes, consisting of 10 boron molecules, to the liposome membranes. This allowed for the liposome cavity to remain free to carry chemotherapeutic compounds. To assess the synergistic effects of chemotherapy and BNCT, a PARP inhibitor was encapsulated within the boronsome. Tumor cell death in vivo was significantly improved following treatment with PARP-encapsulated boronsome-based BNCT in comparison to BNCT without the chemotherapeutic agent. This study suggested that combining DNA repair inhibitors with BNCT can improve the therapeutic impact of BNCT. The prevention of single-strand repair by PARP inhibition likely causes the generation of many DSBs. As prior studies have demonstrated that BNCT-induced DNA damage is processed via the homologous recombination pathway, it is possible that PARP inhibition overwhelms this DSB repair pathway following BNCT, causing increased cell death.

## 5. Immunogenic Potential of BNCT

Although immune effects can be generated by low-LET radiation, clinically relevant immune responses following X-ray irradiation are typically weak, and do not result in robust immunogenic cell death within tumors. Additionally, although abscopal effects (defined as the response of metastatic lesions located far from the field of radiation) have been reported, the data regarding this concept remain sparse [24]. The ability of various anti-cancer drugs and radiation modalities to induce immunogenic cell death is dependent on their ability to generate adequate reactive oxygen species and endoplastic reticulum stress, both of which lead to the activation of immune-associated danger-signaling pathways. The radiobiology of densely ionizing high-LET radiation therapy such as carbon ion therapy and BNCT differs substantially from that of X-ray radiation, and these high-LET therapies are hypothesized to generate significant non-target effects. It is predicted that the highly focal and dramatic oxidative damage and the increased endoplastic reticulum stress generated by high-LET radiation are likely to produce greater immunomodulatory impacts on tumors [25,61,62]. Given this, there is currently immense interest in gaining a better understating of the immunological impacts of high-LET radiation and how these modalities can be synergized with various immunomodulatory drugs to improve cancer outcomes. In this section, we will discuss the immunogenic potential of BNCT, the available data to date, and future directions.

### 5.1. Va: Radiation-Induced Immune Response Pathways

It is important to note that in contrast to our previously held understanding, responses to ionizing radiation are not simply reactionary changes mediated by the generation of DNA strand breaks leading to mitotic catastrophe and cell death. Radiation responses are instead complex and highly dynamic. Mounting evidence to date suggests that through reshaping of the tumor microenvironment, ionizing radiation can awaken the immune system within the host, and enhance the systemic anti-tumor immune response [24,63]. A key driver of this is the cyclic GMP-AMP synthase–stimulator of interferon genes (cGAS-STING) pathway [64]. Ionizing radiation leads to genomic DNA damage and disordered repair, resulting in the formation of cytoplasmic micronuclei containing DNA with asymmetrical chromosomal aberrations. The double-stranded DNA within these micronuclei is enveloped by a fragile micronuclear envelope allowing access to cytoplasmic sensors, evolutionarily conserved mechanisms designed to detect viral RNA and DNA outside the cellular nucleus and mount a robust immune response to counter this insult. Simultaneously, radiation also damages mitochondrial DNA. The presence of cytoplasmic and mitochondrial DNA is detected by cyclic GMP–AMP synthase (cGAS). Conformal changes that occur in cGAS following DNA binding activate its enzymatic capabilities, leading to the synthesis of the second messenger, 2′,3′-cyclic GMP–AMP (cGAMP). STING is an endoplastic reticulum adapter and has been identified as a major modulator of innate immunity. cGAMP generated following cGAS activation binds to STING, leading to the formation of STING tetramers. This structural change leads to coat protein (COPII)-dependent translocation of the STING complex to the Golgi apparatus. The translocated STING then acts as a platform for the recruitment of downstream signaling molecules such as TANK-binding kinase signaling (TBK1). Phosphorylation of the C-terminal tail of STING by TBK1 causes STING activation and allows it to serve as a bridge between TBK1 and interferon regulatory factor 3 (IRF3) kinases. IRF3 is then able to move into the nucleus and facilitate the transcription of type I interferons. Type I interferons induced by the cGAS-STING pathway facilitate the maturation of dendritic cells, migration of dendritic cells to regional lymph nodes, and clonal expansion of cytotoxic T-lymphocytes in regional lymph nodes. These mechanisms facilitate the recruitment of cytotoxic T-lymphocytes to the tumor to incite immunogenic cancer cell death [64,65,66].

The concept of converting an “immunologically cold” tumor to an “immunologically hot” entity is viewed as a potentially revolutionary concept given that it may allow for combining radiation with immunotherapy to improve patient outcomes in otherwise difficult-to-treat tumors. Given the tremendous potential of the cGAS-STING pathway for triggering interferon synthesis and priming T-cells as described above, the impact of BNCT on this pathway must be investigated to uncover means to enhance the immunogenicity of “cold” tumors. Although the impact of BNCT on cytosolic DNA generation and micronuclei formation has not yet been investigated to our knowledge, multiple studies to date have demonstrated increased micronuclei production with high-LET radiation. For instance, in an experiment involving Chinese hamster V79 cells exposed to carbon ion radiation and Cobalt 60 gamma rays, high-LET carbon ions led to significant increases in micronuclei production and DNA ladder formation [67]. Given these findings, one may postulate that the high-LET alpha particles generated via BNCT are also likely to cause similar increases in micronuclei formation and robust activation of the cGAS pathway.

It is important to note, however, that under conditions of photon radiation, the level of cytosolic DNA increases with increasing radiation doses up to 12 to 18 Gy, but then dramatically decreases at higher doses. This is thought to occur due to the metabolism of cytosolic DNA via the DNA exonuclease Trex1 at higher radiation doses [65,68,69]. Balancing the generation of micronuclei formation to elicit cGAS-STING pathway activation while limiting Trex-1 activation remains a key issue. During any future explorations of the impact of BNCT on cGAS-STING pathway activation, it is crucial to determine the minimum threshold radiation doses necessary to activate this pathway while limiting Trex-1 stimulation. Furthermore, research into whether synthetic inhibition of Trex-1 can enhance cGAS signaling following BNCT therapy should also be pursued.

Radiation-induced cell death can cause dying cancer cells to elicit danger-associated molecular patterns (DAMPs), such as calreticulin translocation to the tumor cell surface and the extracellular secretion of ATP and high-mobility group box 1 (HMGB1). These molecules promote the uptake of tumor antigens by dendritic cells and assist in dendritic cell maturation. HMGB1 and the mitochondrial transcription factor A (TFAM) assemble DNA in an optimal fashion through the creation of parallel DNA strands which facilitates the binding of cGAS to DNA [64]. Although these mechanisms have not been investigated in BNCT directly, interestingly, HMGB1 has been reported to play the role of an immune mediator following other alpha particle irradiation. The immunostimulatory potential of alpha particles was investigated in both in vitro and in vivo settings in a study involving MC38 murine colorectal adenocarcinoma cells and murine-tumor-bearing mice. The authors noted that cell cultures treated with the alpha emitter bismuth-213 (^213^Bi) induced the release of DAMPs such as heat shock protein 70 and HMGB1. Immunocompetent mice engrafted with murine tumors were then vaccinated with the alpha-particle-irradiated MC38 cells. Vaccination of mice with irradiated cells led to a significantly superior overall survival in comparison to the unvaccinated cohort (88% vs. 16% at 73 days), suggesting that the alpha-particle-irradiated tumor cells were highly immunogenic and were capable of inducing an in vivo anti-tumor response [70]. Findings from this work are promising and suggest that the alpha particles generated through BNCT have the potential to generate systemic immune effects. The immunogenic potential of BNCT through cGAS-STING pathway activation is demonstrated in Figure 3.

### 5.2. IVb: Existing Data Regarding the Immunogenicity of BNCT and the Potential for Combining with Immunotherapies

Given the potentially immunogenic nature of BNCT, there exists considerable promise for BNCT to be combined with immunotherapies to enhance cancer treatment outcomes. One of the earliest studies investigating such combination therapy was performed in gliosarcoma rat models. In this study, tumor outcomes were assessed in response to BNCT alone vs. BNCT in conjunction with tumor-cell-based immunoprophylaxis. The experimental group of animals received BNCT followed by immunoprophylaxis through subcutaneous injection of unirradiated autologous tumor cells to the left thigh. The control group received BNCT alone. The authors found that the group of animals treated with BNCT and immediate immunoprophylaxis demonstrated increased survival in comparison to the control group. The surviving animals were then rechallenged with implantation of gliosarcoma cells in the contralateral brain one year after initial treatment. Following the rechallenge, the BNCT alone group demonstrated partial immunological protection with 30% of animals surviving at one year, while the experimental group which received BNCT and immunoprophylaxis upfront demonstrated 100% survival [71]. Despite the unconventional means of immunoprophylaxis used in this study, it is suggested that the immunogenicity of BNCT may be synergized with immunomodulatory agents to improve tumor control in aggressive malignancies.

Prior studies have shown that ionizing radiation can morphologically alter macrophages to acquire a more pro-inflammatory phenotype [72]. In a recent preclinical study investigating intravenously delivered liposomal BPA-based BNCT in murine mammary tumor models, boron-rich liposomes preferentially accumulated in peripheral blood mononuclear cells over other blood cell types. Additionally, delivery of neutron irradiation following boron liposome uptake by peripheral blood mononuclear cells caused macrophage polarization, modifying them into an antitumor phenotype and contributing to tumor growth inhibition. This study suggests that liposomal-based boron delivery may increase its immunomodulatory impacts through the alteration of macrophage phenotypes [73].

Trivillan and colleagues examined the potential for abscopal effect generation using BNCT in an experimental rat model of colon cancer. In this study, colon cancer tumors were generated in the right flank of 26 immunocompetent rats. The tumors were then treated with BPA-based BNCT. The control group consisted of 12 animals with unirradiated right flank tumors. Two weeks following BNCT, similar-sized tumors were again generated in the left flank of animals in both experimental and control groups. In comparison to those within the control group, there was a statistically significant reduction in the size of the contralateral untreated tumors among animals within the experimental group. Additionally, when compared to animals with minimal BNCT responses at the irradiated tumor sites, those demonstrating dramatic regression of the irradiated tumors also showed significantly greater size reductions in the untreated contralateral tumors. Although preliminary, this work suggested for the first time that BNCT has the potential to elicit an abscopal effect [26]. The same authors then went on to determine whether the BNCT-induced abscopal effect may be synergized with immunotherapy to further enhance its anti-tumor effects. In this subsequent study, BNCT was combined with Bacillus Calmette-Guerin (BCG) as the immunotherapy agent, and five different experimental groups were generated: BNCT only, BNCT + BCG, BCG only, neutron beam only, and neutron beam with BCG. They found that the BNCT and BNCT + BCG groups demonstrated similar dramatic local anti-tumor responses, while the BCG alone and BCG + BNCT groups demonstrated significant abscopal tumor regression in the contralateral hind leg tumors. Interestingly, the BNCT + BCG group showed significantly less metastatic tumor spread to the regional nodes. This work implied that combining BNCT with immunotherapy has the potential to induce considerable abscopal effects and limit metastatic tumor spread [27].

As blocking CD47 (an integrin-associated transmembrane protein that serves as a ‘don’t eat me’ signal for macrophages and dendritic cells) can stimulate the innate immune system, Chen et al. assessed the potential for combining BNCT with CD47 inhibition as a means of improving tumor response in a mouse glioma model. Given that BNCT reactions occurring in close proximity to DNA increases the potential for dense DNA damage, the authors first generated a nanoparticle liposome boron delivery system known as DOX-CB@lipo-pDNA-iRGD, achieved by combining doxorubicin as the nuclear targeting agent, carborane as the boron compound, and iRGD as the cell-penetrating peptide [74]. CRISPR-Cas9 gene editing technology was then used to knock out the CD47 gene in several glioma cell lines. A mouse glioma model was then developed by the implantation of syngeneic GL261 glioma cells within the brains of immunocompetent C57BL/6 mice. The nucleus-directed DOX-CB@lipo-pDNA-iRGD nanoliposomes were confirmed to deliver boron successfully to these brain tumors, and neutron irradiation was performed. Control groups included tumor-bearing animals treated with BSH + BNCT, CD47 knockdown alone, and neutron irradiation alone. The experimental group treated with BNCT in combination with CD47 knockdown demonstrated significantly higher survival in comparison to the control groups. These data suggested that combining boron delivery with chemo-immunotherapy improves outcomes following BNCT. Furthermore, the expression of cancer stem cell markers such as CD133 and Nestin, which are strongly associated with tumor recurrence and invasion, was significantly downregulated following nanoliposome-based BNCT. In contrast, the BSH-BNCT group did not demonstrate a decrease in these cancer stem cell markers. These findings imply that utilizing a nucleus-directed nanoliposome boron delivery system can elicit a greater immunogenic and anti-cancer stem cell response in comparison to traditional boron agents such as BSH, highlighting the need for developing novel drugs to improve BNCT treatment outcomes. It is also important to note that CD47 is overexpressed in many types of cancer, including ovarian, brain, and lung, as well as in cancer stem cells. Given the positive results noted when CD47 inhibition was combined with nuclear tropic boron chemo liposomes, it may be beneficial to consider all kinds of immune targets beyond just immune checkpoint inhibitors when exploring the synergistic effects of BNCT with immunotherapy.

In an attempt to create the ideal boron delivery agent that is both tumor-specific and highly immunogenic, Hirase and colleagues generated boron-carrying exosomes compatible with BNCT [75]. Exosomes are extracellular vesicles generated inside cells via membrane budding and encapsulate cytosolic molecules. They are secreted through body fluids, can function as second messengers, and may be engineered to serve as unique drug delivery agents. A primary advantage of utilizing exosomes for drug delivery is that they express CD47, which inhibits macrophage–monocyte-based phagocytosis of the exosomes. Inhibition of phagocytosis dramatically increases the circulation time of exosomes within the body and maximizes the potential for boron delivery to tumors [76]. Additionally, exosome-based boron agents are particularly attractive given that exosomes can be specifically targeted to cancer cells via surface molecule modification and can provoke controlled immune responses that may be exploited to increase therapeutic gain [9]. Here, the authors describe a method of encapsulating BSH into exosomes using electroporation. In order to improve the probability of macropinocytosis-induced uptake of the boron-containing exosomes by tumor cells, they also incorporated arginine-rich cell-penetrating peptides into the exosome surface. The presence of arginine-rich attachments on the exosome surface increased cellular boron accumulation and improved cancer cell death following neutron irradiation. With further development of this technology, tumor-derived self-exosomes may be isolated from patients and engineered to carry boron to the tumors in the manner discussed above. As tumor-derived exosomes have various tumor-associated antigens and molecules such as HSP-70/90 and MHC-1 that can trigger an enhanced dendritic-cell- and NK-cell-mediated immune response, it may be possible to exploit the exosome-mediated immune response by combining BNCT with immunotherapy [77].

## 6. The Potential for Combining CAR-T and BNCT

Chimeric antigen receptor T-cell (CAR-T) therapy is approved for the treatment of acute lymphoblastic leukemia, non-Hodgkin lymphoma, and multiple myeloma. It is a cellular therapy that genetically engineers CAR-T cells to attack cancer cells harboring a specific tumor antigen of interest. The revolutionary outcomes of CAR-T therapy for hematological malignancies have triggered multiple clinical trials to investigate its use for solid malignancies. However, CAR-T therapy for solid malignancies has several challenges. Solid tumors, unlike hematological malignancies, express tumor antigen heterogeneity [78]. This leads to malignant cell antigen diversity that hinders identifying a universal tumor-specific antigen for the CAR-T cells to attack. Trafficking and infiltration of CAR-T cells into the tumor tissue is another challenge. Solid tumors express a dense fibrotic matrix, have altered tumor vasculature, and lack the growth factors and chemokines needed to facilitate tumor infiltration [79]. Moreover, the immunosuppressive tumor microenvironment and its associated immunoregulatory cells, cytokines, and chemokines decrease the penetration of CAR-T cells into the tumor [80].

Radiation therapy (RT) can help mitigate the aforementioned challenges of CAR-T therapy in solid malignancies. RT can sensitize tumor cells to immune rejection by overexpressing tumor neoantigens for CAR-T targeting and reducing antigen-negative tumor relapse [81]. RT can increase the local expression of multiple cytokines, which could chemoattract CAR-T cells [82]. RT induces the expression of adhesion molecules on the endothelium which may promote normalization of the tumor vasculature, facilitating homing and transmigration of CAR-T cells [83,84]. RT may induce immunogenic cell death via the elaboration of DAMPs that facilitate the recruitment of antigen-presenting cells (APCs) into the tumor [85,86,87,88]. Lastly, RT can increase target antigen cross-presentation by MHC class II on APCs [89]. Collectively, RT can be used to prime solid tumors for augmented responses to CAR-T therapy.

The promising role of RT in mitigating the challenges of CAR-T therapy for solid malignancies may be more pronounced with high-LET therapies including BNCT. As noted previously, the generation of complex DNA damage and consequent increase in cell death as well as immune-mediated cell death can synergize effectively with CAR-T therapy on multiple levels. First, the increase in cytotoxicity also increases the elaboration of innate tumor-associated antigens and potentially some radiation-induced neoantigens. These are ripe targets for CAR-T therapy since they are only expressed by irradiated cancer cells. Second, the lesser need for oxygen to ‘fix’ the damage caused by high-LET radiation makes these therapies more efficacious at eradicating traditionally radioresistant hypoxic cancer cells. Elimination of these cells provides another opportunity for CAR-T cells to home in to deep recesses of the tumor that harbor recalcitrant resistant cells that thrive in a hostile tumor microenvironment. Third, a greater pro-inflammatory immune microenvironment within the tumor stroma may provide a greater likelihood of cytokine- and chemokine-mediated chemotaxis on CAR-T cells to high-LET irradiated tumors. Lastly, CAR-T cells may be utilized as a vehicle to selectively deliver boron to tumor cells, enhancing the selectivity and efficacy of BNCT. In agreement with these theoretical lines of evidence for synergy between BNCT and CAR-T therapy, a case report showed the combination of BNCT and CAR-T therapy for multifocal glioblastoma to be safe and effective in regressing all intracranial lesions [90]. Taken together, these observations and scientific rationales support the notion that high-LET therapy such as BNCT may be a promising synergistic treatment with CAR-T for solid malignancies and is worth further investigation.

## 7. Conclusions

As clinical evidence of the efficacy of BNCT has evolved in the last decade, so too has our understanding of the molecular mechanisms of anti-tumor efficacy of current BNCT approaches. The time is ripe now to fuel the quest for the next generation of BNCT therapies. This review outlines the potential for next-generation boronated compounds to dramatically increase boron delivery to tumors while sparing normal tissues and underscores the prospect of even greater systemic immune responses being generated when BNCT is optimally coupled with immunotherapy. By summarizing the literature on drug delivery methods that have been explored already and immunotherapy combinations that have been partially explored with BNCT and more extensively investigated with other forms of radiation therapy, this review provides a framework for basing future endeavors to improve BNCT through improved drug development and delivery on one hand, and enhanced generation of local and systemic immune responses on the other hand. Coupled with the increasing availability of accelerator-based BNCT platforms that obviate the need for reliance on reactor-based BNCT facilities, we envision an optimistic outlook where novel BNCT paradigms can advance seamlessly from the bench to the bedside in the foreseeable future.

## Figures and Tables

**Figure 1 cells-12-01398-f001:**
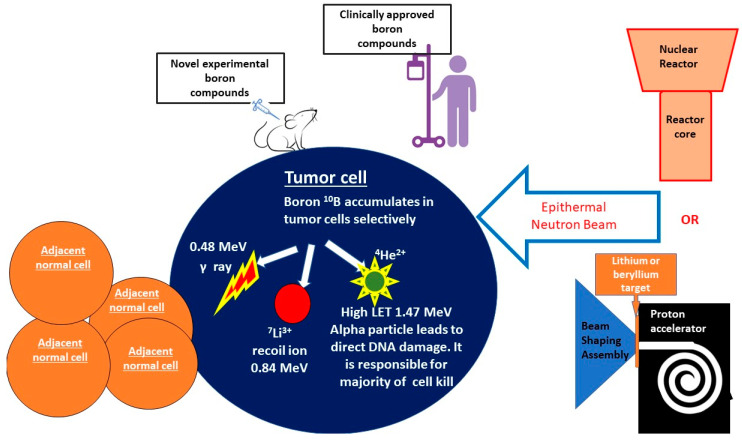
A schematic overview of BNCT. BNCT involves the selective delivery of boronated compounds into tumor cells, followed by exposure of the tumor to epithermal neutron irradiation. Epithermal neutrons maybe generated through the interaction of protons with a lithium or beryllium target, a linear accelerator, or via nuclear reactors. Boron-10 atoms then react with neutrons via a nuclear capture and fission reaction, leading to the production of a high-LET, low-energy, alpha particle and a recoil lithium-7 atom (^10^B + ^1^n → [^11^B]* → ^4^He + ^7^Li) [1,2,3]. The high-LET alpha particles form dense ionization tracks along cellular macromolecules, leading to tumor cell death. As there is limited boron accumulation in normal cells, they are spared from the impacts of high-LET radiation.

**Figure 2 cells-12-01398-f002:**
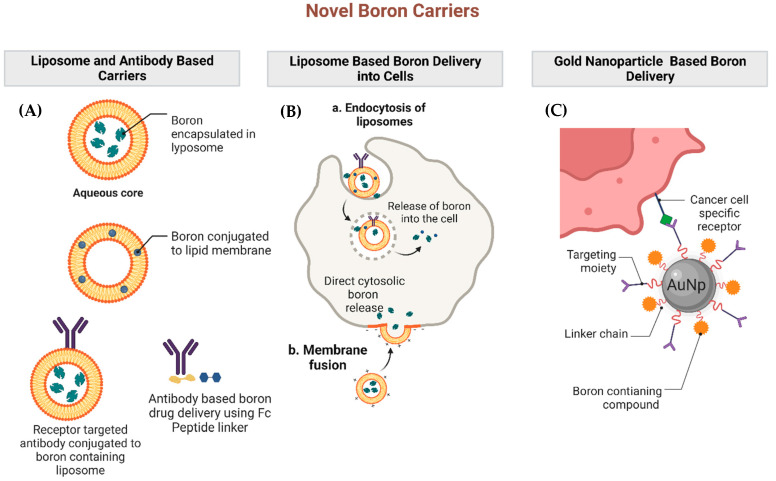
Examples of novel boron carriers and their mechanisms of boron delivery. Several novel classes of boron compounds have been developed and tested in preclinical studies. Panel (**A**) demonstrates liposome and antibody-based boron carriers. Boron may be encapsulated within the lipid bilayer or conjugated to the lipid membrane. For tumor-targeting purposes, a receptor-targeting antibody may be coupled to the boron-containing liposome, or boron may be directly attached to tumor-directed antibodies via peptide linkers. Panel (**B**) demonstrates the mechanism of liposome engulfment and boron delivery to the tumor cells. Panel (**C**) demonstrates nanoparticle-based boron delivery in which boron may be linked to nanoparticles carrying tumor-specific targeting moieties.

**Figure 3 cells-12-01398-f003:**
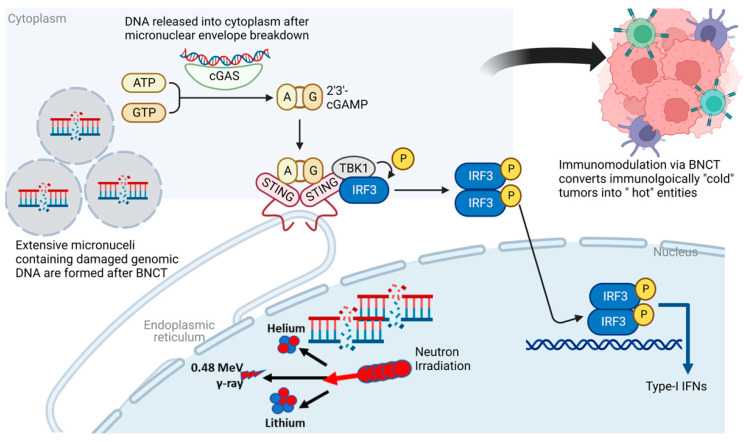
The immunogenic potential of BNCT through cGAS-STING pathway activation. High-LET alpha particles generated through BNCT may cause extensive direct and irreparable DNA damage and the formation of cytoplasmic micronuclei. The presence of cytoplasmic DNA is detected by cGAS, leading to synthesis of the second messenger, cGAMP, which binds to STING, an endoplastic reticulum adapter. STING complex translocates to the Golgi apparatus and recruits TBK1. This leads to phosphorylation of IRF3, which then moves into the nucleus and directs the transcription of type I interferons. Type I interferons facilitate multiple immunomodulatory effects, including the maturation of dendritic cells, migration of dendritic cells to regional lymph nodes, and clonal expansion of cytotoxic T-lymphocytes in regional lymph nodes, ultimately converting an immunologically “cold” tumor into a “hot” entity.

## Data Availability

Not applicable.

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
