# Peer review of "Next-Generation Boron Drugs and Rational Translational Studies Driving the Revival of BNCT"

_cells, 2023, doi:10.3390/cells12101398_

Round 1
Reviewer 1 Report (Previous Reviewer 2)
Seneviratne et al. have revised and resubmitted the review on “Next-generation boron drugs and rational translational studies driving the revival of BNCT.”
In the revised version, in Chapter I, the authors added information from the history of BNCT and the current situation, describing the latest achievements in this unique technology globally and its approval for clinical use in Japan.
The authors have revised and redrawn Figure 1 as previously suggested. The schematic overview of BNCT has become more rational and updated.
The authors have added information on BSH and BPA, which is crucial when describing other novel boron compounds.
Information about dose prediction and calculation has also been added.
In Chapter III, more data on modern boron nanocarriers have been added.
In their response, the authors stated that they had reorganized the article; however, it is totally unclear when looking at the revised version. Nevertheless, they added essential information on critical topics, making the review more updated and coherent.
From this point of view, it is possible to recommend the article for publication.Author Response
Thank you for your thoughtful revision of this manuscript. We appreciate your kind comments.
Reviewer 2 Report (Previous Reviewer 1)
This version improves the previous one, and for me it is suitable for publication, just with a minor suggestion: for being a review where most work on the lines covered should be mentioned, I miss key references in addition to [14-16] for the important topic of the improvement in BNCT dosimetry and RBE determinations (even some published in this same journal)
Author Response
Thank you for your thoughtful revision of this manuscript. We have now added additional new references regarding this topic.
Reviewer 3 Report (New Reviewer)
This is a very well-written comprehensive review not only for boron delivery agents but also entire BNCT clinical and research fields. Especially this reviewer has been impressed the new findings of BNCT and tumor-immunology including CAR-T technology.
However, this reviewer suggests the authors to mention more appropriate references than they mentioned in original manuscripts.
For example,
1) Phase II JHN002 study was published originally by Hirose K. et al. in Radiother Oncol 155:182-, 2021. The authors should add this article as an appropriate reference.
2) Also Kawabata and his colleagues have published accelerator-based BNCT phase 2 trial for recurrent GBM in Neuro-Oncol Adv, Add this article in an appropriate section.
3) The authors mentioned Scwint and her colleagues works as original for simultaneous use of 2 deferent boron delivery agents. However, Ono and his colleagues published already the rationale of simultaneous use of BSH and BPA in animal works prior to Schwint’s publication.
Int J Radiat Oncol Biol Phys 34:1081–1086, 1996, ibid 43:431–436, 1999
Also Miyatake et al used these compounds simultaneously for the same case in clinical BNCT.
J Neurosurg 103:1000–1009. Mention these articles in appropriate sections.
4) In recent few years, most impressing publication in BNCT fields in Japan might be Science Adv by Nomoto et al.2020;6:eaaz1772, 2020. They reported the PVA-BPA can be internalized into cancer cells through LAT-1 mediated endocytosis.
Mention this article in LAT-1 section.
For minor revision;
1) In Japan, no reactor-based BNCT is applicable for clinical use, so far.
In Fig.1, move the position of an arrow including “Epithermal Neutron Beam” a little bit lower position. Best position should be just beside “or”.
Author Response
This is a very well-written comprehensive review not only for boron delivery agents but also entire BNCT clinical and research fields. Especially this reviewer has been impressed the new findings of BNCT and tumor-immunology including CAR-T technology.
However, this reviewer suggests the authors to mention more appropriate references than they mentioned in original manuscripts.
Thank you for your thoughtful review. We have now made the changes you have recommended. For example,
- Phase II JHN002 study was published originally by Hirose et al. in Radiother Oncol 155:182-, 2021. The authors should add this article as an appropriate reference.
This reference has been added.
- Also Kawabata and his colleagues have published accelerator-based BNCT phase 2 trial for recurrent GBM in Neuro-Oncol Adv, Add this article in an appropriate
This reference has been added.
- The authors mentioned Scwint and her colleagues works as original for simultaneous use of 2 deferent boron delivery However, Ono and his colleagues published already the rationale of simultaneous use of BSH and BPA in animal works prior to Schwint’s publication.
Int J Radiat Oncol Biol Phys 34:1081–1086, 1996, ibid 43:431–436, 1999
Also Miyatake et al used these compounds simultaneously for the same case in clinical BNCT. J Neurosurg 103:1000–1009. Mention these articles in appropriate sections.
Thank you for this comment. We have added information regarding these publications to the section titled “Translational work to date involving BNCT”.
- In recent few years, most impressing publication in BNCT fields in Japan might be Science Adv by Nomoto et al.2020;6:eaaz1772, 2020. They reported the PVA-BPA can be internalized into cancer cells through LAT-1 mediated endocytosis.
Mention this article in LAT-1 section.
Thank you for this comment. We have now added information from the publication into the manuscript.
For minor revision;
1) In Japan, no reactor-based BNCT is applicable for clinical use, so far.
In Fig.1, move the position of an arrow including “Epithermal Neutron Beam” a little bit lower position. Best position should be just beside “or”.
Thank you for this comment. We have now altered the figure as recommended.
We confirm that this manuscript is not currently under consideration for publication elsewhere. We hope that the updated version is satisfactory to the reviewers.
We appreciate the opportunity to submit this work to your journal for consideration of publication.
Please address all correspondence concerning this manuscript to Dr. Seneviratne at seneviratne.danushka@mayo.edu
Sincerely,
Danushka Seneviratne, MD, PhD
This manuscript is a resubmission of an earlier submission. The following is a list of the peer review reports and author responses from that submission.
Round 1
Reviewer 1 Report
This is a very useful, well-written review on the subject, that covers topics not previously well describes, as the immunogenic aspects of BNCT.
Therefore I recommend publication, although some minor suggestions follow:
Line 67. Tha authors say "BSH has shown some success in preclinical rat glioma models and in the treatment of recurrent head and neck cancers." That is not accurate. BSH has shown sucess in a number of clinical trials of GBM and other brain tumors in Japan and Finland with the use of neutrons from research reactors.
Line 99. The authors say "... remains an area of active investigation [12]", but they only provide a single reference which corresponds to a microdosimetry model. I suggest they include some recent paper about experimental measurements of the CBE.
Lines 107-109: "However, given the growing national and global interest in improving the therapeutic efficacy of oncologic therapies, preclinical and clinical investigations related to BNCT have significantly expanded over the last decade." It might be interesting to add that the current growing interest on BNCT is also motivated by the new accelerator-based neutron sources, that can be built in hospitals, as opposite to the previous sources, which were research reactors. It is only said at the end of the conclusions.
Line 133. "As neither BPA nor BSH are ideal boron carriers..." It should be interesting to justify this sentence. It is because of the total uptake, the selectivity or both?
Line 288: "carrying 10 boron molecules", do you mean 10 boron atoms per carborane or 10 carboranes in the liposome membrane? Please clarify
Section V. Immunogenic potential of BNCT. A general comment: in my opinion, one of the aspects that makes BNCT and particle therapy better than photon therapy in this sense is that they are superior in the balance between pro-immunogenic and anti-immunogenic effects because they spare much more circulating T-cells. But I leave this to the consideration of the authors.
Reviewer 2 Report
In their review, according to the title, Seneviratne et al. planned to describe next-generation boron drugs for BNCT and translational studies aimed at “the revival of BNCT”. Additionally, the authors aimed to discuss the immunomodulatory potential of BNCT. The review contains several chapters describing different aspects of BNCT. However, it is too early to recommend this review for publication due to the following issues.
1. The general review of BNCT, which is given in chapter I, contains outdated and incorrect information to some extent.
a. There is nothing new in chapter I. All the information has already been published in numerous research articles and reviews. Only “small Phase I/II trials” on BNCT are mentioned. The authors seem to be unaware that in 2020 accelerator-based BNCT was approved for clinical use in Japan and is covered by the national health insurance for adults with head and neck cancer. Nothing is told about the registered boron drug.
b. In the schematic overview of BNCT, the neutron source is schematically shown as a cyclotron, though linear accelerators have also been developed for BNCT. There is no information about nuclear reactors that are still clinically used in certain countries. The neutron beam is shown as the thermal one, which is incorrect, as the epithermal beam is used for clinical BNCT. As an object, a mouse is shown, whereas BNCT is nowadays applied for humans, as it was initially designed. The tumor in the mouse is a single mass, whereas the main designation of BNCT is to treat invasive cancers, where tumor cells are located among healthy cells. It would be better to show an alpha-particle with its energy, rather than Helium, as the particle is not the atom at that stage, but the atomic nucleus carrying a large portion of energy, which is also not disclosed, but quite important compared to the gamma-ray energy, which influence is relatively little.
c. The requirements for BNCT drugs are not fully described.
2. In chapters II and III, only certain selected publications are included in the discussion. Numerous publications on recently developed boron compounds and nanocarriers are not included. Figure 2 is also limited to a small number of the types of born nanocarriers. There is no information about elemental boron nanoparticles, boron carbide and boron nitride nanoparticles, and other carriers that have revolutionized boron retention in the tumor to avoid the drawback of BPA and BSH. The literature review seems to be incomplete. There were several special issues related to BNCT published in Cells and other journals, and using the information from recent publications would help to enrich the review.
3. Chapter IV describes several reactor-based BNCT studies by the Argentinian research group. However, nuclear reactors were not previously mentioned as neutron sources. The epithermal neutron beam has been described for irradiating animals, although only thermal neutrons have been mentioned before.
4. There is no information about human glioma animal models, though gliomas, being invasive cancers, are primary targets of BNCT. And numerous studies using human glioma models have been conducted.
5. The authors tend to include BNCT in the group of other high-LET irradiation-based therapies. Though alpha particles deliver high-density ionization during BNCT, the effect of heavy ions from accelerators on biological tissues may differ due to the different nature of these irradiation types and their influence on tumor and normal tissues from the outside. The unique feature of BNCT is intracellular multicomponent irradiation.
6. Well-established human or animal tumor cell lines were used in all cases described. Due to the relative genetical stability of such cells, the immunomodulating potential of therapies might be more prominent. When we face actual cancers with a high mutation potential, the approaches described by the authors might be even more challenging.
7. Chapter V is the speculation on the possibility of using CAR-T therapy for the tumors that are targets of BNCT. Due to the completely different designs of these anticancer modalities, the ideas described by the authors may remain only theoretical speculations. Gliomas do not metastasize, and multiple tumor masses, as well as recurrences, may differ genetically. Malignant lymphoma, which is often differentiated from glioblastoma in the case of brain tumors, may be presented by multiple tumor masses in the brain. However, it is efficiently treated using chemotherapy without the need for BNCT.
The review was written by several authors and still looks like a disjointed work with relatively weakly linked chapters. It would be better to focus mainly on the potential immunomodulating abilities of BNCT and its combination with immunotherapies in such a review.